# SARS-CoV-2 Dysregulates Neutrophil Degranulation and Reduces Lymphocyte Counts

**DOI:** 10.3390/biomedicines10020382

**Published:** 2022-02-04

**Authors:** Abenaya Muralidharan, Todd A. Wyatt, St Patrick Reid

**Affiliations:** 1Department of Pathology and Microbiology, College of Medicine, University of Nebraska Medical Center, Omaha, NE 68198-5900, USA; abenaya.muralidharan@unmc.edu; 2Department of Environmental, Agricultural & Occupational Health, College of Public Health, University of Nebraska Medical Center, Omaha, NE 68198-5900, USA; twyatt@unmc.edu; 3Veterans Affairs Nebraska-Western Iowa Health Care System, Omaha, NE 68105, USA

**Keywords:** neutrophils, elastase, MPO, degranulation, SARS-CoV-2, lymphocytes

## Abstract

SARS-CoV-2, the virus that causes COVID-19, has given rise to one of the largest pandemics, affecting millions worldwide. High neutrophil-to-lymphocyte ratios have been identified as an important correlate to poor recovery rates in severe COVID-19 patients. However, the mechanisms underlying this clinical outcome and the reasons for its correlation to poor prognosis are unclear. Furthermore, the mechanisms involved in healthy neutrophils acquiring a SARS-CoV-2-mediated detrimental role are yet to be fully understood. In this study, we isolated circulating neutrophils from healthy donors for treatment with supernates from infected epithelial cells and direct infection with SARS-CoV-2 in vitro. Infected epithelial cells induced a dysregulated degranulation of primary granules with a decrease in myeloperoxidase (MPO), but slight increase in neutrophil elastase release. Infection of neutrophils resulted in an impairment of both MPO and elastase release, even though CD16 receptor shedding was upregulated. Importantly, SARS-CoV-2-infected neutrophils had a direct effect on peripheral blood lymphocyte counts, with decreasing numbers of CD19+ B cells, CD8+ T cells, and CD4+ T cells. Together, this study highlights the independent role of neutrophils in contributing to the aberrant immune responses observed during SARS-CoV-2 infection that may be further dysregulated in the presence of other immune cells.

## 1. Introduction

Severe acute respiratory syndrome coronavirus 2 (SARS-CoV-2) is a β coronavirus with a single-stranded RNA genome. It is the third virus in this group to show potential for causing large-scale pandemics, with SARS-CoV and MERS-CoV (Middle East respiratory syndrome coronavirus) causing outbreaks in 2003 and 2012, respectively [1,2,3,4,5]. In March 2020, the World Health Organization (WHO) declared SARS-CoV-2-infection-induced coronavirus disease 2019 (COVID-19) a pandemic. By November 2020, SARS-CoV-2 proved its tremendous infectivity and transmissibility by spreading to 216 countries and territories, infecting over 62 million individuals and killing over 1.5 million [6]. 

Although the majority of the infected individuals are asymptomatic or exhibit mild symptoms, about 15% develop pneumonia [7]. Infection in the higher-risk groups, such as the elderly and individuals with underlying chronic conditions, can cause acute respiratory distress syndrome and multiorgan failure resulting in death [7]. Many of the early symptoms resemble other respiratory viral infections, with high fever and dyspnea being the main difference between COVID-19 and the common cold [8]. Importantly, compared to infections by other commonly circulating viruses such as influenza, SARS-CoV-2 infection has higher chances of progressing to a critical state requiring oxygen therapy and ventilatory support. This suggests SARS-CoV-2 may have a systemic aspect to its infection that is accompanied by severe inflammation [9,10,11,12].

Hyperinflammation has been identified as one of the major causes of the morbidity and mortality observed in COVID-19. Specifically, neutrophilia correlates to COVID-19 disease severity, with increased blood neutrophil counts in severe patients compared to mild cases [13]. This, combined with lymphopenia, leading to elevated neutrophil-to-lymphocyte ratio (NLR), has been observed as a hallmark of severe COVID-19 suggestive of poor recovery rates [7,14,15]. 

Neutrophils make up 50–70% of all leukocytes and are the most abundant immune cells in human blood [16]. They serve as the first responders during infections and can shape cell-mediated responses. Although their role during bacterial and fungal infections have been well-studied, their role during viral infections remains to be fully understood [17,18,19]. Upon activation, neutrophils migrate to a target tissue where they defend against invading microbes. They can also interact with other immune cell populations and affect the microenvironment [18,20]. 

During SARS-CoV-2 infection, whole blood transcriptomics of patients who required intensive care showed increased neutrophil function and activation genes on the first day of hospitalization [21,22]. This indicates that neutrophil activation occurs before the onset of severe illness. Another study looking at 300 patients with confirmed COVID-19 observed that the presence of circulating activated neutrophils can serve as an independent predictor for mechanical ventilation and death [23]. 

Numerous clinical data collected during the pandemic, specifically regarding neutrophils and NLR, have immensely aided patient care in terms of predicting prognosis and devising a more informed treatment plan. However, in-depth mechanistic understanding of the neutrophilic dysregulation observed during SARS-CoV-2 infection is not fully understood. This knowledge is crucial for establishing better treatment strategies to allow for improved control of disease progression. As such, we aimed to systematically delineate the effects of SARS-CoV-2 on neutrophil degranulation and subsequent lymphocyte numbers in vitro. 

Here, using neutrophils from healthy donors, we found a dysregulation in neutrophil degranulation induced by SARS-CoV-2-infected epithelial cells and the virus directly, which was characterized by diminished myeloperoxidase (MPO) release and impaired azurophil granule release following CD16 shedding, respectively. Notably, infected human neutrophils significantly lowered lymphocyte counts, specifically B cells, CD8+ T cells, and CD4+ T cells in vitro. Together, this report sheds light on the mechanisms underlying the aberrant neutrophil activity observed in COVID-19.

## 2. Materials and Methods

### 2.1. Cells and Virus

Calu-3 cells (ATCC: HTB-55) were grown in Eagle’s Minimum Essential Medium (EMEM) supplemented with 10% fetal bovine serum (FBS). Whole blood, collected from healthy donors by University of Nebraska Medical Center (UNMC) Elutriation Core Facility, was used for neutrophil isolation with EasySep™ Human Neutrophil Isolation Kit (StemCell Technologies, Vancouver, Canada) according to manufacturer’s instructions. Peripheral blood lymphocytes (PBLs) were isolated from whole blood by the UNMC Elutriation Core Facility.

SARS-CoV-2 wild-type strain USA-WA1/2020 (BEI Resources, Manassas, VA, USA) was propagated in Calu-3 cells. All experiments were conducted at the UNMC Biosafety Level 3 (BSL3) facility unless the samples were fixed with 4% paraformaldehyde (Thermo Fisher Scientific, Waltham, MA, USA). Fixed samples were further analyzed at Biosafety Level 2 (BSL2).

### 2.2. Neutrophil Treatment with Calu-3-Conditioned Media

Calu-3 cells were infected with SARS-CoV-2 at multiplicity of infection (MOI) of 0.01 and 0.1 for 48 h. Supernatants were collected and UV-inactivated on ice for 15 min. Human neutrophils were seeded at a density of 1 × 10^6^ cells/well in a 96-well round-bottom plate and treated with the UV-inactivated Calu-3-conditioned media. Neutrophils treated with 10 ng/mL phorbol myristate acetate (PMA; InvivoGen, San Diego, CA, USA) were used as controls. Following a 4 h treatment at 37 °C and 5% CO_2_, the plates were centrifuged, and supernatant media were collected for ELISA while the cells were used for immunofluorescence.

### 2.3. SARS-CoV-2 Infection of Neutrophils

Human neutrophils were seeded at a density of 1 × 10^6^ cells/well in a 96-well round-bottom plate and infected with SARS-CoV-2 at MOI of 0.001 and 0.01. Neutrophils were also treated with 10 ng/mL PMA to serve as controls. Two hours post infection, the plates were centrifuged for collection of supernates and cells. Supernatant media were used for ELISA and PBL treatment while cells were used to determine viral titer and for flow cytometry analysis.

### 2.4. PBL Treatment with Neutrophil Conditioned Media and Coculture

Following a 2 h SARS-CoV-2 infection of neutrophils, supernates were collected and UV-inactivated on ice for 15 min. Human PBLs isolated from whole blood of healthy donors were seeded at a density of 1 × 10^6^ cells/well in a 96-well round-bottom plate and treated with the UV-inactivated neutrophil-conditioned media. Conditioned media from neutrophils treated with 10 ng/mL of PMA or 10 ng/mL ultrapure lipopolysaccharide (LPS; InvivoGen, San Diego, CA, USA) were also used for PBL treatment. Untreated, PMA- or LPS-treated PBLs were used as controls. Furthermore, infected and PMA/LPS-treated neutrophils were cocultured with 1 × 10^6^ cells/well PBLs. Following an 18 h treatment at 37 °C and 5% CO_2_, the plates were centrifuged and cells were collected for flow cytometry analysis of PBLs.

### 2.5. RNA Extraction and Quantitative Polymerase Chain Reaction (qPCR)

Cells were scraped and collected in Buffer AVL with carrier RNA (Qiagen, Hilden, Germany). RNA was then isolated from the samples using QIAamp Viral RNA Mini Kit (Qiagen, Hilden, Germany) according to manufacturer’s instructions. RNA was also isolated from a heat-inactivated cell lysate and supernate containing SARS-CoV-2 isolate USA-WAI/2020 (BEI Resources, Manassas, VA, USA) with known genome equivalents of virus, which was used as a standard during qPCR.

UltraPlex 1-Step ToughMix (QuantaBio, Beverly, MA, USA) was used along with 2019-nCoV CDC Probe and Primer Kit for SARS-CoV-2 (Catalog: KIT-nCoV-PP1-1000) for the qRT-PCR reactions. QuantStudio 3 Real-Time PCR machine (Applied Biosystems, Waltham, MA, USA) was used with QuantStudio Design and analysis software version 1.5.1 (Applied Biosystems, Waltham, MA, USA) for analysis. Results are expressed as log of genome equivalents/mL.

### 2.6. Immunofluorescence

Treated neutrophils were fixed with 4% paraformaldehyde and permeabilized with 0.1% Triton X-100 (Sigma-Aldrich, St. Louis, MO, USA). After blocking with 3% bovine serum albumin (BSA), the cells were stained with a rabbit monoclonal recombinant anti-human neutrophil elastase antibody (Abcam, Cambridge, United Kingdom) at 1:250 dilution. The cells were then washed with 1x PBS and stained with Alexa Fluor 488-conjugated goat anti-rabbit IgG (Invitrogen, Waltham, MA, USA) at 1:2000 dilution. The cells were again washed and stained with a mix of Hoechst 33,342 nuclear stain (Invitrogen, Waltham, MA, USA) and CellMask Deep Red plasma membrane stain (Invitrogen, Waltham, MA, USA) at 1:20,000 dilution each. Stained cells were visualized using the Operetta CLS™ system (Perkin Elmer, Waltham, MA, USA). Alexa Fluor 488-positive cells were identified under 40x water objective and 55 fields/well were analyzed using Harmony 4.9 (Perkin Elmer, Waltham, MA, USA) software.

### 2.7. Myeloperoxidase (MPO) and Neutrophil Elastase ELISA

Human MPO ELISA Kit (Abcam, Cambridge, UK) and Human Neutrophil Elastase ELISA Kit (Abcam, Cambridge, UK) were used according to manufacturer’s instructions. Cell supernates were diluted 5-fold before being added to the ELISA plates.

### 2.8. Flow Cytometry

Neutrophils were washed and stained with FITC-conjugated anti-human CD11b (clone ICRF44), PE-conjugated anti-human CD66b (clone G10F5), PE-Cyanine5-conjugated anti-human CD14 (clone 61D3), PE-Cyanine7-conjugated anti-human CD15 (clone HI98), and Alexa Fluor 647-conjugated anti-human CD16 (clone 3G8; Thermo Fisher Scientific, Waltham, MA, USA). Samples containing PBLs were washed and stained with PE-conjugated anti-human CD19 (clone HIB19; BioLegend, San Diego, CA, USA), PE-Cyanine5-conjugated anti-human CD3 (clone UCHT1), Alexa Fluor 488-conjugated anti-human CD8a (clone OKT-8), and Alexa Fluor 647-conjugated anti-human CD4 (clone RPA-T4). All antibodies were purchased from eBioscience (San Diego, CA, USA) unless otherwise specified. Stained samples were fixed with 4% paraformaldehyde. Data were acquired using NovoCyte flow cytometer (ACEA Biosciences, San Diego, CA, USA). Single-stained and unstained controls were used for compensation to correct for spectral overlap. Data analysis was completed using NovoExpress 1.5.0 (Agilent Technologies, Santa Clara, CA, USA) software.

### 2.9. Statistical Analysis

Statistical analysis was conducted using Student’s t-test or one-way analysis of variance (ANOVA) when appropriate. Tukey’s post hoc test was used to adjust for multiple comparisons between different test groups. Tests were performed at a 5% significance level. All statistical analyses were performed using GraphPad Prism 8 (San Diego, CA, USA) software.

## 3. Results

### 3.1. Factors Secreted by SARS-CoV-2-Infected Epithelial Cells Diminish Myeloperoxidase Release While Modestly Increasing Elastase Release by Human Neutrophils In Vitro

To determine the effect of infected lung epithelial cells on neutrophil degranulation in vitro, we infected Calu-3 cells with SARS-CoV-2 at two different MOIs for 48 h. To ensure effective cell entry and virus release, we determined the viral titer in the cells and supernatant using qPCR (Figure 1A). At 48 h post infection, some cytopathic effect was observed at the higher MOI, leading to decreased viral load in the cells and the supernatant compared to MOI 0.01. The supernatant from the infected Calu-3 cells were then UV-irradiated for treatment of circulating human neutrophils isolated from healthy donors.

Following a 4 h treatment with Calu-3-conditioned media, we measured the release of MPO and neutrophil elastase in the supernatant using ELISA. A PMA treatment control was added to serve as a positive control for potent neutrophil activation. Conditioned media from infected Calu-3 cells significantly reduced MPO release compared to the uninfected control media (Figure 1B). However, a dose–response relationship in MPO release with increasing infection was not observed. A MOI higher than 0.1 may be required to detect a significant drop in MPO release. As expected, a low dose of PMA induced high levels of MPO release (Figure 1B).

In contrast, levels of secreted neutrophil elastase slightly increased at the higher MOI compared to the uninfected control (Figure 1E). To determine if this increase correlated to lower intracellular elastase levels, the treated neutrophils were fixed, permeabilized, and stained with an antineutrophil elastase antibody. Lower percentage of elastase-positive cells were indeed seen after treatment with PMA and conditioned media from infected Calu-3 compared to the uninfected control (Figure 1C,D). However, the proportion between intracellular and released elastase seen with PMA treatment was not detected with infected conditioned media treatment (Figure 1C,E). Together, these data suggest that SARS-CoV-2-infected epithelial cells induce aberrant neutrophil degranulation, specifically in the azurophil granules, in vitro.

### 3.2. Direct Infection of Neutrophils with SARS-CoV-2 Promotes CD16 Shedding but Does Not Increase Release of Azurophil Granules

Next, we sought to observe the direct effect of live SARS-CoV-2 on neutrophil degranulation. Circulating neutrophils isolated from healthy donors were infected with live virus for 2 h. The cells were then collected for qPCR to determine the viral load. A dose-dependent response was observed with a 10-fold increase in MOI resulting in a ~10-fold increase in viral genome detected (Figure 2A).

We then analyzed the activation state of the neutrophils by measuring CD16 shedding. CD16, also known as FcγRIII, is the most abundant receptor on the surface of neutrophils [24]. During neutrophil activation, CD16 is shed, granting access to other activating receptors on the cell surface [24]. Using flow cytometry, we determined the percentage of CD16-negative cells among the CD11b+ CD66b+ CD14− CD15+ population. Neutrophils infected at MOI 0.01 had a significant increase in CD16 shedding compared to uninfected controls (Figure 2B). As expected, stimulation with PMA resulted in the highest percentage of CD16 cells, indicating the highest activation state.

Since CD16 shedding is associated with enhanced degranulation [24,25], we quantified secreted MPO and elastase in the supernatant 2 h post infection. Interestingly, infection significantly lowered the levels of released MPO (Figure 2C) and neutrophil elastase (Figure 2D). While both MOI 0.001 and 0.01 resulted in reduced MPO release, only the higher MOI decreased elastase release. It is important to note that there is some MPO and elastase release even with very low CD16 shedding as seen in the ‘No virus’ control, suggesting that the neutrophils are not at resting state in ex vivo conditions, as anticipated. Moreover, as expected from the high CD16 shedding, a low-dose PMA treatment induced effective MPO and elastase release, albeit to the same levels as the uninfected control (Figure 2C,D).

### 3.3. SARS-CoV-2-Infected Neutrophils Reduce B Cell, CD8+ T Cell, and CD4+ T Cell Counts In Vitro

As part of contributing to the inflammatory milieu, the release of granules by neutrophils has a significant effect on lymphocyte numbers, and in turn, on their function [26]. Therefore, we ascertained the effect of infected neutrophils on B- and T-cell populations in vitro. Peripheral blood lymphocytes (PBLs) were isolated from healthy donors and either treated directly with PMA or LPS, or with UV-irradiated conditioned media from neutrophils treated with live virus, PMA or LPS two hours post-treatment. The PBLs were also co-cultured with treated neutrophils. Following an 18 h treatment, the PBLs were analyzed using flow cytometry. The number of CD3− CD19+ B cells (Figure 3A), CD19− CD3+ CD8+ T cells (Figure 3C), and CD19− CD3+ CD4+ T cells (Figure 3E) significantly decreased with infected conditioned media, while media from LPS-treated neutrophils had comparable numbers to the uninfected control. A similar trend was observed when treated neutrophils were co-cultured with PBLs (Figure 3B,D,F,G).

Since no other stimulants were added to induce active proliferation of these lymphocytes, we did not expect any differences between ‘No virus’ and LPS-conditioned media or co-culture groups for B- and T-cell populations. Furthermore, PBLs either directly treated with PMA or treated with PMA-stimulated neutrophils were drastically diminished in number. This may be due to a possible toxic effect of PMA on lymphocytes.

The lack of other stimulants also resulted in lower B- and T-cell counts in the untreated (PBLs only) group than the uninfected neutrophil-conditioned media treatment. Similarly, direct LPS treatment of PBLs resulted in decreased lymphocyte numbers compared to treatment with media from LPS-stimulated neutrophils. This may be due to the higher expression of Toll-like receptor 4 (TLR4) on neutrophils than unstimulated PBLs [27,28]. Higher TLR4 levels leads to higher LPS-induced activation of neutrophils that can further release lymphocyte-activating factors. In addition, lymphocytes upregulate TLR4 expression upon CD3/CD28 activation [28]. Since there were no such activators present, direct LPS treatment of PBLs did not augment the lymphocyte counts (Figure 3).

Although infection of neutrophils had a significant impact on lymphocyte numbers, an increase in MOI only had a mild effect during neutrophil-conditioned media treatment. However, when the PBLs were co-cultured with the infected neutrophils, a substantial decrease in CD8+ (Figure 3D) and CD4+ (Figure 3F) T cells was seen with increasing MOI, even though the overall counts were lower compared to conditioned media treatment. This may be due to additional effects neutrophils can have on lymphocytes through direct contact. Together, these data highlight the significant role neutrophils have on B- and T-cell counts during SARS-CoV-2 infection.

## 4. Discussion

Many comorbidities have been associated with mortality in COVID-19 patients. One of the major risk factors for mortality is coinfections, specifically secondary bacterial pneumonia [29]. Other viral pandemics have proven the detrimental effect of bacterial coinfections on viral diseases. A recent study comparing COVID-19 and influenza patients showed that higher rates of bacterial infections were present in COVID-19 patients [29]. Interestingly, these bacterial infections were more common in fatal cases [30,31]. Indeed, the highest percentage of bacterial coinfections were found in critically ill COVID-19 patients compared to moderately or severely ill patients [32,33]. Colonization of bacteria is thought to be augmented by dysregulated virus-induced immune responses, and neutrophils are known to play a crucial role in controlling bacterial infections [34,35]. In this study, we show the adverse effects of SARS-CoV-2 on neutrophil functions, which may contribute to the lethality caused by secondary bacterial lung infections in severe COVID-19 cases.

Neutrophilia and altered neutrophil function have been identified as hallmarks of immunopathology associated with severe COVID-19 [36]. Elevated levels of neutrophils have been observed in the nasopharyngeal epithelium, distal parts of the lung, and in the blood [37,38]. Notably, blood transcriptomes of severe patients showed an increase in neutrophil activation markers, and immunophenotyping revealed the presence of immature neutrophil subsets in the blood [22,39,40,41]. Similarly, many studies have identified neutrophil signatures and/or conducted ex vivo characterization of neutrophils isolated from infected patients. However, the alterations occurring in previously healthy neutrophils during early SARS-CoV-2 infection remains unclear. Here, we report for the first time the dysregulated degranulation, specifically the release of MPO and elastase, in healthy human neutrophils as a result of direct infection and due to secreted factors from infected epithelial cells in vitro. Furthermore, we show the direct effect of SARS-CoV-2-infected neutrophils on lymphocyte numbers.

A unique feature of SARS-CoV-2 was described in mice, which was not observed with other RNA or DNA viruses, where an increase in infiltration of immature aberrant neutrophils was observed in the lungs, correlating to a fatal outcome [42]. Patients with severe COVID-19 were also shown to have high levels of immature and low-density neutrophils [39,40,43,44]. In MERS-CoV and SARS-CoV infections, persistently activated neutrophils helped maintain the inflammatory state by releasing cytokines in the lungs [45]. Likewise, with SARS-CoV-2, neutrophils were reported to have a shift towards their immature forms and to display enhanced degranulation of primary (azurophil) granules and proinflammatory cytokine release, thereby driving hyperinflammation [46]. This study by Parackova et al. varies from our study in that they used peripheral blood neutrophils from COVID-19 patients to study neutrophil function and properties, whereas we used circulating neutrophils from healthy donors to observe the effect of an in vitro SARS-CoV-2 infection on neutrophil function. Furthermore, in their experiments, Parackova et al. found increased MPO and neutrophil elastase in the serum of COVID-19 patients [46]. Since, in our experiments, other immune cells were absent during infection of healthy neutrophils and we used a set number of cells compared to the neutrophilia seen during COVID-19, we observed a virus-mediated decrease in MPO and elastase release (Figure 2).

On the other hand, CD16 shedding typically indicates an activated state of neutrophils, but increased shedding did not translate to increased release of MPO and elastase in our study (Figure 2). This aberrant activation may lead to longer lifespan of neutrophils persistently contributing to the inflammation. Indeed, early elevation of developing and mature neutrophil counts in COVID-19 patients were predictive of higher mortality rates [22]. Upregulation of neutrophil genes and chemokines was also observed in bronchoalveolar lavage fluid cells from COVID-19 patients [47].

Molecules contained in neutrophil granules can promote tissue damage in addition to the injury caused by invading pathogens. Molecules such as MPO and elastase are also components of neutrophil extracellular traps (NETs), and help modulate immune responses [48]. During SARS-CoV-2 infection of healthy human neutrophils, NET release was observed in a MOI-dependent manner that subsequently promoted epithelial cell death in vitro [49]. Furthermore, sera from COVID-19 patients induced NET release by healthy human neutrophils in vitro [50]. This suggests that virus-mediated release of granules as part of NETs or through degranulation may have different effects on the virus and surrounding tissue.

Moreover, a thorough comparative analysis COVID-19 blood transcriptomes and over 3100 samples derived from 12 different viral infections and inflammatory diseases revealed highly specific signatures for COVID-19 [41]. Interestingly, single-cell analysis of circulating cells showed a disease-stage-dependent downregulation of MHC molecules on monocytes and granulocytes, and immune cell exhaustion [39,40,51,52]. Along with higher gene expression of neutrophil elastase (ELANE) and MPO, as expected due to the neutrophilia, upregulation of T-cell suppressor genes, such as IL-10 and PD-L1, was observed [41]. The transcriptome had evidence of simultaneous inflammatory and suppressive markers, highlighting a dysregulated phenotype in the peripheral granulocytes [41].

The downregulation of MHC molecules on granulocytes and upregulation of T-cell suppressor genes seen in COVID-19 patients coincide with our observation of reduced lymphocyte counts following treatment with SARS-CoV-2-infected neutrophils (Figure 3). Neutrophils have been reported to enter lymph nodes during bacterial infections and to interact with B cells, affecting antibody production and thus the humoral response [53]. Furthermore, neutrophils release H_2_O_2_ that can suppress T-cell proliferation and activation through various mechanisms such as inducing apoptosis [54,55]. MPO catalyzes H_2_O_2_ to form reactive oxygen intermediates that play an important role in neutralizing microbes [26,48]. Therefore, lower levels of MPO release, as observed in our study, induced by SARS-CoV-2-infected neutrophils (Figure 2) may lead to higher levels of hydrogen peroxide resulting in reduced CD4+ and CD8+ T-cell counts (Figure 3). Indeed, high NLR has been implicated with severe COVID-19 and is associated with poor prognosis [10,56,57,58].

One of the main limitations of our study is the lack of access to lung neutrophils, as immune processes in the lungs may differ compared to peripheral blood. However, circulating immune cells can provide helpful information regarding the overall inflammatory state. In summary, we have shown dysregulation of neutrophil responses, specifically in degranulation, mediated directly by SARS-CoV-2 and SARS-CoV-2-infected epithelial cells in the absence of other immune cells. Clinical data suggest that this aberrance is further augmented when other cells are present, contributing to the inflammatory milieu. Therefore, it is crucial to dissect helpful versus harmful immune responses in COVID-19 and to understand the mechanisms underlying the hyperinflammation observed in severe cases in order to develop better treatment strategies. Further studies using neutrophil-specific inhibitors or activators can help deduce specific pathways involved in protection and potentially provide the basis for development of novel therapeutic options.

## Figures and Tables

**Figure 1 biomedicines-10-00382-f001:**
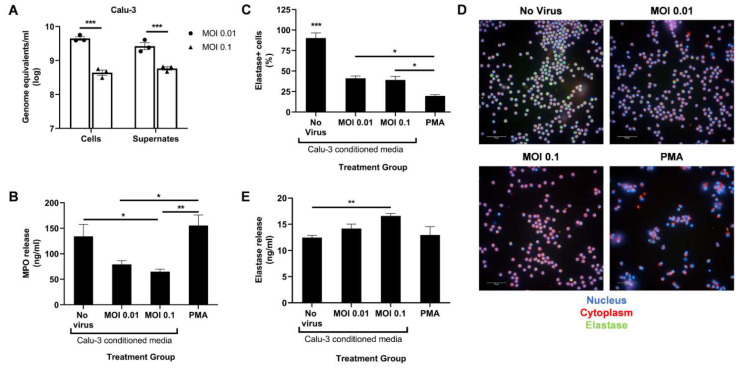
SARS-CoV-2-infected Calu-3 cells secrete factors that diminish neutrophil MPO release. (**A**) Viral titer in the cells and supernates of SARS-CoV-2-infected Calu-3 48 h post infection; UV-irradiated Calu-3 supernates, referred to as Calu-3-conditioned media, were used to treat human neutrophils. (**B**) MPO levels in the treated neutrophils supernatant media determined through ELISA; (**C**) Percentage of neutrophil elastase-positive cells among total cells; (**D**) Representative images at 40× magnification of neutrophils stained for their nucleus (blue), cytoplasm (red), and neutrophil elastase (green); (**E**) Neutrophil elastase levels in the treated neutrophils supernatant media determined through ELISA. Data shown are mean ± SEM; *n* = 3 per group in each experiment per donor; experiments were repeated twice with two different donors; * *p* < 0.05, ** *p* < 0.01, *** *p* < 0.001 (one-way ANOVA with Tukey’s post hoc test).

**Figure 2 biomedicines-10-00382-f002:**
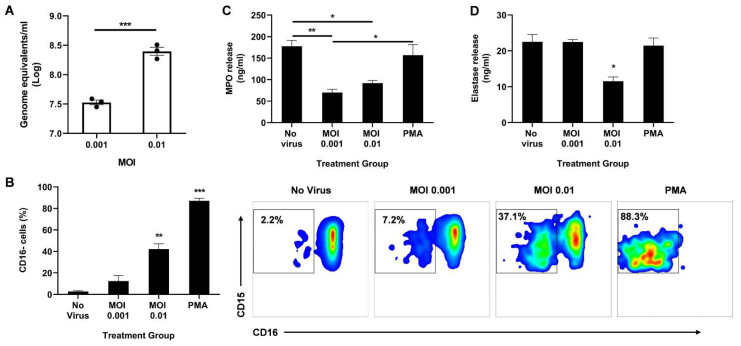
Direct infection of neutrophils with SARS-CoV-2 promotes CD16 shedding without increasing MPO or elastase release. (**A**) Viral titer in the neutrophils 2 h post infection; (**B**) Percentage of CD16 cells among CD11b+ CD66b+ CD14− CD15+ population (left). Representative flow cytometry density plots of CD11b+ CD66b+ CD14− CD15+ CD16− cells in different treatment groups (right); Levels of MPO (**C**) and neutrophil elastase (**D**) in the supernatant media determined through ELISA. Data shown are mean ± SEM; *n* = 3 per group in each experiment per donor; experiments were repeated three times with three different donors; * *p* < 0.05, ** *p* < 0.01, *** *p* < 0.001 (Student’s t-test or one-way ANOVA with Tukey’s post hoc test).

**Figure 3 biomedicines-10-00382-f003:**
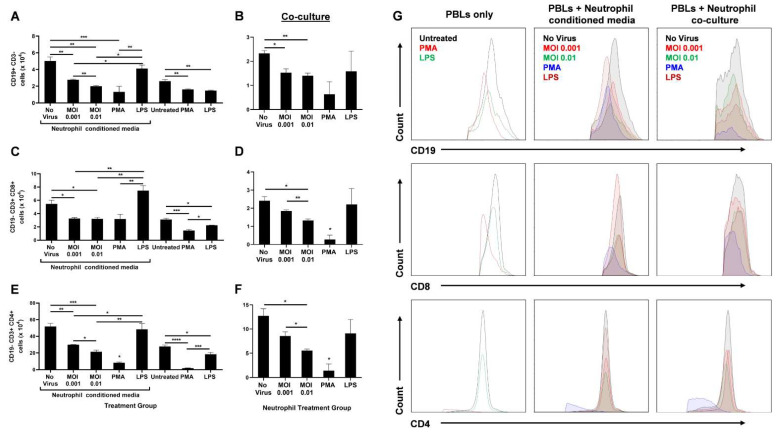
SARS-CoV-2-infected neutrophils reduce lymphocyte counts in vitro. After 18 h treatment of PBLs with conditioned media from infected neutrophils or direct stimulation with PMA or LPS, the number of CD19+ CD3- B cells (**A**), CD19− CD3+ CD8+ T cells (**C**), and CD19− CD3+ CD4+ T cells are shown (**E**). Following 18 h coculture of PBLs with infected neutrophils, the number of CD19+ CD3− B cells (**B**), CD19− CD3+ CD8+ T cells (**D**), and CD19− CD3+ CD4+ T cells are shown (**F**); (**G**) Representative flow cytometry histograms of B and T cells in different treatment groups of PBLs alone, PBLs treated with neutrophil-conditioned media, and PBL/neutrophil co-culture. Data shown are mean ± SEM; *n* = 3 per group in each experiment per donor; experiments were repeated three times with three different donors; * *p* < 0.05, ** *p* < 0.01, *** *p* < 0.001 (one-way ANOVA with Tukey’s post hoc test).

## Data Availability

The data presented in this study are available on request from the corresponding author.

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
