# Peer review of "SARS-CoV-2 Dysregulates Neutrophil Degranulation and Reduces Lymphocyte Counts"

_biomedicines, 2022, doi:10.3390/biomedicines10020382_

Round 1
Reviewer 1 Report
The Abstract should be more focused. The description of other clinical sudies and methods should be removed.
The Introduction is too long. Some statements are too simple, eg. about neutrophil subpopulation.
Author Response
Reviewer #1
The Abstract should be more focused. The description of other clinical studies and methods should be removed.
Response:
We thank the reviewer for the comment. We have removed the description of other studies from the abstract.
The Introduction is too long. Some statements are too simple, eg. about neutrophil subpopulation.
Response:
Thank you for the comment. We have shortened the introduction.
Reviewer 2 Report
Very interesting article, however, the following has to be cleared:
Page 3. Materials and Methods: Human neutrophils were seeded at a density of 1x106 cells/well in a 96-well round bottom plate and infected with SARS-CoV-2 at MOI of 0.001 and 0.01.
However, in Figure 1 you have MOI 0.01 and MOI 0.1. Then in Figure 2, MOI 0.01 and 0.001.
Confused, what MOIs did you use? Please correct or re-phrase to be clear.
Figure 2B: again left panel, MOI 0.001 and 0.01, right panel 0.01 and 0.1.
Also, from how many healthy subjects did you obtain the PMNs and PBLs?
How many repetitions of each experiment? You state n= 3 per group in each experiment per donor; experiments were repeated with two different donors. Please be more specific.
Minor: Please spell out abbreviations upon first use.
Author Response
Reviewer #2
Very interesting article, however, the following has to be cleared:
Page 3. Materials and Methods: Human neutrophils were seeded at a density of 1x106 cells/well in a 96-well round bottom plate and infected with SARS-CoV-2 at MOI of 0.001 and 0.01. However, in Figure 1 you have MOI 0.01 and MOI 0.1. Then in Figure 2, MOI 0.01 and 0.001. Confused, what MOIs did you use? Please correct or re-phrase to be clear.
Response:
We thank the reviewer for the comment and apologize for the confusion. In Figure 1, Calu-3 cells were infected at MOI 0.01 and 0.1, and the UV-irradiated supernatant was used for neutrophil treatment, while in Figure 2, neutrophils were directly infected at MOI of 0.001 and 0.01. We have altered Figure 1 caption to clarify this:
“Figure 1. SARS-CoV-2 infected Calu-3 cells secrete factors that diminish neutrophil MPO release. (A) Viral titer in the cells and supernates of SARS-CoV-2 infected Calu-3 48 hours post-infection; UV-irradiated Calu-3 supernates, referred to as Calu-3 conditioned media, were used to treat human neutrophils. (B) MPO levels in the treated neutrophils supernatant media determined through ELISA; (C) Percentage of neutrophil elastase positive cells among total cells; (D) Representative images at 40x magnification of neutrophils stained for their nucleus (blue), cytoplasm (red), and neutrophil elastase (green); (E) Neutrophil elastase levels in the treated neutrophils supernatant media determined through ELISA.”
Figure 2B: again left panel, MOI 0.001 and 0.01, right panel 0.01 and 0.1.
Response:
We apologize for this typo. Headings on Figure 2B right panel have been corrected to match the left panel.
Also, from how many healthy subjects did you obtain the PMNs and PBLs? How many repetitions of each experiment? You state n= 3 per group in each experiment per donor; experiments were repeated with two different donors. Please be more specific.
Response:
Thank you for the comment. We have clarified the numbers in each figure caption.
Minor: Please spell out abbreviations upon first use.
Response:
We have edited the manuscript to spell out all abbreviations upon first use.
Round 2
Reviewer 2 Report
I thank the authors for their response